# Extracellular Vesicles in Synovial Fluid from Rheumatoid Arthritis Patients Contain miRNAs with Capacity to Modulate Inflammation

**DOI:** 10.3390/ijms22094910

**Published:** 2021-05-06

**Authors:** Andrew D. Foers, Alexandra L. Garnham, Simon Chatfield, Gordon K. Smyth, Lesley Cheng, Andrew F. Hill, Ian P. Wicks, Ken C. Pang

**Affiliations:** 1The Walter and Eliza Hall Institute of Medical Research, Parkville 3052, Australia; andrew.foers@kennedy.ox.ac.uk (A.D.F.); garnham.a@wehi.edu.au (A.L.G.); simon.chatfield@mh.org.au (S.C.); smyth@wehi.edu.au (G.K.S.); 2Department of Medical Biology, University of Melbourne, Parkville 3052, Australia; 3Department of Rheumatology, Royal Melbourne Hospital, Parkville 3050, Australia; 4School of Mathematics & Statistics, University of Melbourne, Parkville 3010, Australia; 5Department of Biochemistry and Genetics, La Trobe Institute for Molecular Science, La Trobe University, Bundoora 3086, Australia; L.Cheng@latrobe.edu.au (L.C.); Andrew.Hill@latrobe.edu.au (A.F.H.); 6Murdoch Children’s Research Institute, Parkville 3052, Australia; 7Department of Paediatrics, University of Melbourne, Parkville 3052, Australia; 8Department of Adolescent Medicine, Royal Children’s Hospital, Parkville 3052, Australia

**Keywords:** rheumatoid arthritis, extracellular vesicles, miRNA, synovial fluid

## Abstract

In rheumatoid arthritis (RA), extracellular vesicles (EVs) are associated with both the propagation and attenuation of joint inflammation and destruction. However, the specific EV content responsible for these processes is largely unknown. Investigations into identifying EV content are confounded by the challenges in obtaining high-quality EV preparations from synovial fluid. Implementing a size exclusion chromatography-based method of EV isolation, coupled with small RNA sequencing, we accurately characterised EV miRNAs in synovial fluid obtained from RA patients and investigated the differences between joints with high- and low-grade inflammation. Synovial fluid was obtained from the joints of 12 RA patients and, based on leukocyte counts, classified as either high (*n* = 7)- or low (*n* = 5)-grade inflammation. Using size exclusion chromatography, EVs were purified and small RNA was extracted and sequenced on a NextSeq 500. Sequencing reads were aligned to miRBase v21, and differences in miRNA profiles between RA patients with high- and low-grade joint inflammation were analysed. In total, 1972 distinct miRNAs were identified from RA synovial fluid EVs. miRNAs with less than five reads in fewer than five patients were filtered out, leaving 318 miRNAs for analysis. Analysis of the most abundant miRNAs suggested that they negatively regulate multiple genes relevant to inflammation, including signal transducer and activator of transcription 3 (STAT3), which lies downstream of IL-6 and has a pro-inflammatory role in RA. Synovial fluid from joints with high-grade inflammation contained 3.5-fold more EV miRNA per mL of synovial fluid (*p* = 0.0017). Seventy-eight EV miRNAs were differentially expressed between RA joints with high- and low-grade inflammation, and pathway analysis revealed that their target genes were commonly involved a variety of processes, including cellular apoptosis, proliferation and migration. Of the 49 miRNAs that were elevated in joints with high-grade inflammation, pathway analysis revealed that genes involved in cytokine-mediated signalling pathways were significantly enriched targets. In contrast, genes associated with reactive oxygen species signalling were significantly enriched as targets of the 29 miRNAs elevated in joints with low-grade inflammation. Our study identified an abundance of EV miRNAs from the synovial fluid of RA patients with the potential to modulate inflammation. In doing so, we defined potential mechanisms by which synovial fluid EVs may contribute to RA pathophysiology.

## 1. Introduction

Rheumatoid arthritis (RA) is a chronic systemic autoimmune disease that targets synovial joints. RA causes chronic, progressive joint inflammation with exacerbations and remissions. Unless it is adequately treated, disease persistence leads to irreversible joint destruction and subsequent joint deformity, resulting in progressive disability. In RA, joint damage occurs due to an immune response that drives inflammatory cell infiltration. The pathophysiology of RA is incompletely understood. To improve patient outcomes and guide the development of more effective ways to control this disease, a better understanding of the disease processes is required.

Extracellular vesicles (EVs)—including exosomes, microvesicles and apoptotic bodies—are phospholipid bilayer-bound vesicles that are released from host cells and which contain a bioactive cargo of RNAs, proteins and lipids [1]. In RA, synovial fluid (SF) EVs have immunomodulatory functionality and are described to both contribute to and protect against joint destruction [2,3].

MicroRNAs (miRNAs) are ~18–25 nucleotide RNAs that can regulate protein translation through complementary binding to mRNA transcripts [4]. In RA, fluctuations in the expression of miRNAs may contribute to the severity of joint inflammation and destruction. For example, elevated miR-146a levels are reported in RA SF [5] and in CD4+ T cells isolated from RA SF [6], but not in osteoarthritis SF, whereas miR-146a expression in SF CD4+ T cells is positively correlated with SF TNF levels [6]. A direct role for miR-146a in promoting inflammation through upregulating T cell activity is supported by miR-146a overexpression, suppressing apoptosis in Jurkat T-like cells [6].

An increasing number of studies have also demonstrated that miRNAs encapsulated within EVs can promote inflammatory disease. For instance, miRNAs in Treg-derived EVs supressed T helper 1 cell proliferation and pro-inflammatory cytokine release [7]. Conversely, GU-rich motifs present in some EV miRNAs have been described to act as endolysosomal TLR8 (TLR7 in mice) agonists [8]. Consistent with this, EVs in the sera of HIV patients were found to contain GU-rich viral miRNAs that act as ligands for TLR8 and stimulate TNF release in recipient macrophages [9].

However, much remains to be discovered about the contributions of SF EV miRNAs to RA pathophysiology. In this study, we therefore profiled EV miRNAs within SF from a cohort of RA patients with either high- or low-grade inflammation and explored the mechanisms by which EV miRNAs might contribute to or protect against joint destruction in RA.

## 2. Results

### 2.1. Characterisation of EV Isolation

EVs were first prepared by means of size exclusion chromatography (SEC) from SF obtained from a cohort of RA patients with either high- or low-grade inflammation (Table 1 and Appendix A). To confirm that EV preparations were of satisfactory quality, EV enrichments were assessed by Western blotting for the canonical EV markers: HSP70, syntenin, FLOT1, Rab 27b, TSG101 and annexin 1 (Figure 1A). Transmission electron microscopy confirmed the presence of EVs with minimal apparent non-EV contaminating material (Figure 1B). These observations are consistent with our previous report detailing SEC as a method for obtaining high-quality EV enrichments from SF [10]. Previously, we demonstrated that RA patients with high-grade synovial inflammation have increased concentrations of SF EVs [11]. Consistent with this, we detected a 3.5-fold increase in the EV miRNA concentration per mL of SF in joints with high-grade inflammation (*p*-value = 0.0017; Figure 1C), suggesting the potential involvement of SF EV miRNAs in inflammatory processes in RA.

### 2.2. Highly Ranked SF EV miRNAs Target Immunomodulatory SF EV Proteins

To characterise the miRNA present in the SF of RA patients, small RNA sequencing was performed. A total of 1415 miRNA species were identified and, after filtering out lowly abundant species, 318 miRNAs remained. The 10 highest-ranked miRNAs across the entire patient cohort are specified in Table 2. To investigate biological processes associated with these prevalent miRNAs, experimentally validated gene targets were identified using miRTarBase (Appendix A) and biological processes associated with the gene targets were characterised (Figure 2). A robust enrichment for genes involved in regulating gene expression was observed, including AKT1 (miR-100-5p, miR-10b-5p and miR-99a-5p), FOXO1 (let-7a-5p and miR-21-5p), KRAS (let-7a-5p), IKBKB (miR-148a-3p), STAT3 (let-7a-5p, miR-148a-3p, miR-21-5p and miR-92a-3p), TGFBR1 (let-7b-5p) and TP53 (miR-10b-5p). In addition, genes associated with cytokine-mediated signalling were targets of highly prevalent miRNA, including the inflammatory mediators CCR1 (miR-21-5p), IL-1β (miR-21-5p), IL-6 (let-7a-5p and miR-26a-5p), MCL1 (miR-26a-5p), NOS2 (miR-26a-5p), PIK3CA (miR-10b-5p) and SOCS5 (miR-92a-3p).

To investigate if these 10 highly-ranked miRNAs might synergise to repress the expression of individual genes, common gene targets were identified. Genes targeted by four or more miRNA are listed in Table 3. Interestingly, the Akt activator insulin-like growth factor 1 receptor (IGF1R) is a target of five of the 10 highest-ranked miRNAs, and the pro-inflammatory transcription factor STAT3 is targeted by four of the 10 highest-ranked miRNAs.

### 2.3. Seventy-Eight SF EV miRNAs Are Differentially Expressed between RA Patients with High- and Low-Grade Inflammation

To investigate how SF EV miRNAs might more specifically contribute to RA pathophysiology, differences in miRNA expression in RA joints with either high- or low-grade inflammation were characterised. Overall, 78 differentially expressed miRNAs were defined. Of these, 49 were elevated in high-grade inflammation, whereas 29 were elevated in RA joints with low-grade inflammation (Figure 3A and Table 4). Results for all miRNAs are presented in Appendix A.

To further investigate miRNA contributions within joints with high-grade inflammation, gene targets of the 49 miRNAs that increased in highly inflamed joints were identified and the associated biological processes were explored. Pathway analysis again revealed an enrichment for genes associated with processes regulating gene expression (Figure 3B). ‘Positive regulation of transcription by RNA polymerase II’ was the highest-ranked biological process, with 22% of all target genes associated. The ‘cytokine mediated signalling pathway’ was also again highly ranked, with target genes including AKT1 (miR-143-3p, miR-185-3p and miR-192-5p), CCL20 (miR-21-5p), CCL3 (miR-223-3p), CCR1 (miR-21-5p), CISH (miR-150-5p), CSF1R (miR-155-5p), CXCL8 (miR-155-5p), FASLG (miR-21-3p and miR-21-5p), IL-1β (miR-21-5p), IL6R (miR-221-5p), JAK2 (miR-101-3p), MMP9 (miR-143-3p), STAT1 (miR-150-5p, miR-155-5p and miR-223-3p), STAT3 (miR-21-3p, miR-21-5p and miR-223-3p), TGFB1 (miR-185-5p and miR-21-5p), TNF (miR-143-3p) and TP53 (miR-150-3p, miR-150-5p, miR-25-3p and miR-28-3p) (Appendix A). Some of the 49 miRNAs found to be increased in EVs from highly inflamed joints may synergistically repress the translation of genes involved in driving inflammatory processes, as demonstrated by multiple miRNAs targeting proteins typically associated with promoting inflammation, including IGF1R, PTEN, VEGFA and BCL2 (Table 5).

Finally, the functionality of the 29 miRNAs found to be increased in EVs within the joints of RA patients with low-grade inflammation were investigated. Pathway analysis of gene targets revealed an enrichment for genes associated with apoptotic processes, cell migration, gene expression and cell proliferation (Figure 3C). ‘Negative regulation of apoptotic processes’ was the highest-ranked biological process, with miRNA target genes including AKT1 (miR-451a), CD44 (miR-328-3p), IGF1R (miR-214-3p, miR-23b-3p, miR-486-5p and miR-92b-3p), IKBKB (miR-451a), IL6 (miR-451a), MIF (miR-1228-5p and miR-451a) and MYC (miR-451a) (Appendix A). Interestingly, similarly to the above observations in joints with high-grade inflammation, multiple miRNAs found to be enriched in joints with low-grade inflammation again target PTEN and TP53 (Table 6).

## 3. Discussion

In this study, SF EVs from RA patients were found to contain a cargo of miRNAs with the capacity to modulate inflammation. Highly abundant miRNAs, as well as those found to be enriched in patients with either high- or low-grade joint inflammation, were observed to target genes involved in similar or identical biological pathways. Many of the highly prevalent and differentially expressed miRNAs in both disease groups target genes associated with the positive regulation of transcription. For example, PTEN is targeted by multiple miRNAs enriched in high- and low-grade inflammation, as well as four of the 10 most prevalent miRNAs across all patients. Considering our previous observation of significant increases in total SF EV numbers within RA joints with high-grade inflammation when compared to those with low-grade inflammation [11]—combined with the significant enrichment in total EV miRNA concentration from joints with high-grade inflammation seen here (3.5-fold; *p*-value = 0.0017)—EV miRNA levels within RA joints could have a profound effect on inflammatory activity, whereas differences in EV miRNA profiles between joints with high- and low-grade inflammation might make more subtle contributions.

Highly-ranked EV miRNAs were found to share common inflammatory gene targets, suggesting coordinated, synergistic involvement in downregulating inflammation. In particular, five of the 10 highest-ranked miRNAs are regulators of the inflammatory driver IGF1R. IGF1R contributes to RA progression though Akt activation and JAK/STAT signalling [12], and has shown promise as a potential therapeutic target in RA [13]. STAT3 was similarly targeted by four of the 10 highest-ranked miRNAs and is a potent positive regulator of inflammation that has also shown promise as a potential therapeutic target in RA [14,15]. In this way, SF EV miRNAs could represent a negative feedback mechanism that prevents excessive inflammation.

A number of specific EV miRNAs were identified through differential expression analysis and these may have important modulatory effects on joint inflammation in RA. For instance, miR-451a was enriched 15-fold (adj. *p*-value = 1.9 × 10^−3^) in SF EVs from RA joints with low-grade inflammation and has been previously described to suppress inflammation by inhibiting: (i) activation of the Akt/mTOR pathway [16], (ii) cytokine expression [17] and (iii) T-cell activation [18]. As another example, miR-223-3p was enriched 12-fold (adj. *p*-value = 6.4 × 10^−4^) in EV enrichments from the joints of RA patients with high-grade inflammation and has been described as both a promoter of and protector against inflammatory disease. On the one hand, miR-223-3p promotes osteoclast differentiation and activation [19,20] and, consistently with this, the silencing of miR-223-3p reduces the severity of collagen-induced arthritis in mice [21]. On the other hand, downregulation of miR-223-3p increases IL-1β and IL-6 expression and STAT3 signalling in macrophages [22], and miR-223-3p knockout mice spontaneously develop inflammatory lung disease [23]. Although superficially contradictory, these results are in keeping with the fact that individual miRNAs have a diverse array of target transcripts and—depending on the cell type in which they are expressed—may function to both promote and protect against disease [24].

Pathway and target analyses between differentially expressed miRNAs revealed functional similarities between the gene targets of SF EV miRNAs enriched in the joints of RA patients with either high- or -low-grade inflammation. In order to accurately identify miRNA function, our study focused on experimentally validated miRNA gene targets. However, this approach is biased towards well-studied genes. This includes PTEN, which has received considerable interest due its function as a key tumour suppressor. In our dataset, PTEN was identified as a target of miRNAs that were enriched in both RA subgroups. Although this indicates regulatory roles for SF EV miRNAs in regulating PTEN expression, miRNA gene targets with potentially greater pathophysiological relevance might not have been identified as they are poorly characterised. Notably, several of the miRNAs that were among the most differentially expressed in our study have received little attention in the past. Of these, miR-4508 had the highest statistical significance of all the miRNAs investigated and was enriched 12-fold (adj. *p*-value = 3.9 × 10^−4^) in EVs from joints with low-grade inflammation. Additionally, miR-1228-5p was enriched 15-fold (adj. *p*-value = 2.1 × 10^−3^) in EVs from joints with low-grade inflammation, and miR-3614-5p (adj. *p*-value = 1.9 × 10^−3^) and miR-1976 (adj. *p*-value = 1.2 × 10^−3^) were similarly both enriched 13-fold in EVs from joints with high-grade inflammation. Future studies to delineate the potential roles of these miRNAs in RA pathogenesis may therefore be illuminating.

Notably, several abundant and enriched SF EV miRNAs identified in this study might also stimulate pro-inflammatory responses in RA by acting as TLR8 ligands. Of the 10 highest-ranked miRNAs, miR-21-5p [25,26], let7a [27] and let7b [28] are all reported to activate TLR8. Likewise, miR-21-5p [25,26] and miR-142-3p [29] were both significantly enriched in EVs from joints with high-grade inflammation (adj. *p*-values = 2.2 × 10^−2^ and 1.6 × 10^−2^, respectively) and are described as endolysosomal TLR agonists. Other EV miRNAs identified in this study may also have the capacity to activate TLR8, as indicated by their elevated GU content. Consistent with previous observations that miRNAs containing 3′ GU motifs stimulate inflammatory responses by activating TLR8 [25], miR-7-5p, which was enriched 3-fold (adj. *p*-value = 3.6 × 10^−2^) in EVs from joints with high-grade inflammation, and miR-1180-3p, which was enriched 4-fold (adj. *p*-value = 8.5 × 10^−3^) in EVs from joints with low-grade inflammation, both have >90% GU content in their 3′ halves. Although further studies are required, these data suggest that miRNA-mediated TLR8 activation might be an unappreciated mechanism by which SF EVs promote joint inflammation in RA.

To our knowledge, this is the first study to comprehensively characterise and compare differences in SF EV miRNAs between the joints of RA patients with high- and low-grade inflammation. Using high-quality EV purifications from SF, we have described a diverse miRNA cargo within EVs that suggests combined roles for SF EVs in RA, in both protecting against and driving joint inflammation and destruction. Although our analysis was limited by a small cohort, we defined a panel of miRNAs that are likely involved in immunomodulation. Investigating the functionality of SF EVs within the RA synovium is a priority for future work, including identifying recipient cell types and characterising miRNA targets. Looking forward, the delivery of anti-inflammatory miRNAs via EVs and/or EV-mimic liposomes might present new therapeutic options for the treatment of RA.

## 4. Materials and Methods

We have submitted all relevant data from our experiments to the EV-TRACK knowledgebase (EV-TRACK ID: EV210057) [30].

### 4.1. Patient Details, Collection and Storage of Human Synovial Fluid

SF was obtained from RA patients undergoing arthrocentesis, as previously described [10,11], and used with informed consent and the approval of the Melbourne Health Human Research Ethics Committee (projects 2005.056 and 2010.293). Following needle aspiration, SF was centrifuged at 2000× *g* for 20 min to remove cells, then aliquoted and stored at −80 °C until the time of experiment. SF was classified as originating from joints with either high- or low-grade inflammation, as reported previously [11]. Specifically, the inflammatory status of the index aspirated joint was assessed based on the presence of SF white cell counts of either greater or less than 2000 cells/µL, which is a cut-off previously assessed as showing reasonable performance in distinguishing between inflammatory and non-inflammatory disease (sensitivity: 0.84; specificity: 0.84) [31]. Patient demographics and clinical parameters are summarised in Table 1, with individual patient details provided in Appendix A.

### 4.2. Sample Preparation and EV Isolation

EVs were isolated using size exclusion chromatography (SEC), as previously described [10]. A total of 2–5 mL of cell-depleted SF was thawed, treated with hyaluronidase (Sigma-Aldrich, Macquarie Park, NSW, Australia) at 30 U/mL and DNase I (Worthington, Biochemical, Lakewood, CA, USA) at 20 U/mL for 15 min at 37 °C. Enzyme-treated, cell-depleted SF was diluted to 13 mL with 4.84 mM EDTA/DPBS and centrifuged at 10,000× *g* (avg) (11,700 RPM, k-Factor = 1563) for 30 min at 4 °C in a 70 Ti rotor using polycarbonate tubes (Beckman Coulter, Mount Waverley, VIC, Australia). The supernatant was collected and injected into a HiPrep 26/60 Sephacryl S-500 HR prepacked gel filtration column and eluted with 4.84 mM EDTA/DPBS at a flow rate of 1.5 mL/min. SEC eluent from 6–120 min was collected, transferred to polycarbonate tubes and ultracentrifuged at 100,000× *g* (avg) (36,900 RPM, k-factor = 157) in a 70 Ti rotor for 90 min at 4 °C to concentrate EVs. EV pellets were resuspended in 250 µL of DPBS and stored at −80 °C until the time of RNA extraction.

### 4.3. Transmission Electron Microscopy

EVs were prepared as above and pellets were fixed overnight at 4 °C with 1% glutaraldehyde and adsorbed onto glow-discharged 200-mesh formvar/carbon coated Cu grids (ProSciTech, Kirwan, QLD, Australia). Grids were washed twice with milliQ water, negatively stained with 2% uranyl acetate and imaged using a Talos L120C electron microscope (FEI, Hillsboro, OR, USA).

### 4.4. Western Blot Analysis

Gel electrophoresis and Western blotting were performed as previously described [10]. Primary antibody details are specified in Appendix A.

### 4.5. Enrichment and Separation of Small RNA

EVs in PBS were thawed on ice and combined with 750 µL of Trizol LS (ThermoFisher, Scoresby, VIC, Australia). RNA was extracted using the miRNeasy mini kit (Qiagen, Chadstone, VIC, Australia) in accordance with the manufacturer’s instructions and following the Appendix A protocol to separately enrich for small RNA using the RNeasy MinElute Cleanup Kit (Qiagen). miRNA concentrations were assessed with an Agilent 2100 Bioanalyser, using Small RNA chips (Agilent, Mulgrave, VIC, Australia).

### 4.6. miRNA Library Preparation and Sequencing

cDNA libraries were prepared and uniquely barcoded using the NEB Next Multiplex Small RNA Library Prep Sets 1 for Illumina (NEB, Ipswich, MA, USA) in accordance with the manufacturer’s instructions, using adapter ligation incubation conditions of 18 h at 16 °C and reagents diluted for low RNA input, as instructed by the manufacturer. To capture the cDNA-miRNA and deplete adapter dimers and excess primers, size selection was performed using Agencourt AMPure XP Beads (Beckman Coulter) according to the NEB supplementary protocol but with cDNA-miRNA captured at a bead concentration of 1.3X. Enriched cDNA libraries with unique barcodes were pooled and sequenced with a NextSeq 500 (Illumina, Scoresby, VIC, Australia) using the 75 cycle High Output kit v2 (Illumina) to generate paired-end 75 base reads.

### 4.7. Bioinformatic Analysis of Sequencing Data

RNA sequencing reads were analysed using R Statistical software version 3.4.3. For analysis of miRNA, Illumina adapter sequences were removed from small RNAseq reads with TrimGalore and directly aligned to mature miRNA sequences defined in miRBase v21 using the Subread aligner [32]. The number of reads uniquely mapping to each miRNA were counted using featureCounts [33]. Counts were normalised using the trimmed mean of M values method. The total counts and normalisation factors are specified in Appendix A. A total of 1972 miRNA species were identified. Unfiltered counts of all identified miRNA are specified in Appendix A. miRNAs with less than 5 reads in fewer than 5 patients were removed, leaving 318 miRNA species for analysis. Sequence data were fit into a negative binomial generalised log-linear model with patient sex incorporated into the design matrix using edgeR [34]. Differential expression analysis was performed with edgeR and *p*-values were adjusted using the Benjamini–Hochberg procedure. miRNAs with an adjusted *p*-value < 0.05 were considered differentially expressed, which has the effect of controlling the false discovery rate below 5%. Raw fastq files have been uploaded to the Sequence Read Archive (PRJNA687167).

### 4.8. Ranking of Prevalent miRNAs

To identify prevalent miRNAs within SF EVs, miRNAs were ranked in individual patients according to their expression levels. For each miRNA, an average rank across all patients was calculated. miRNAs were then given a final ranking according to the average rank. This method was selected to avoid undue influence of outliers and differentially expressed miRNAs, that would confound average expression counts.

### 4.9. Pathway Analysis

Pathway analyses were performed with FunRich v3.1.3 using the UniProt database [35]. Gene targets of miRNAs that have been experimentally observed by either Western blotting or the reported assay were identified using miRTarBase 7.0 [36].

## Figures and Tables

**Figure 1 ijms-22-04910-f001:**
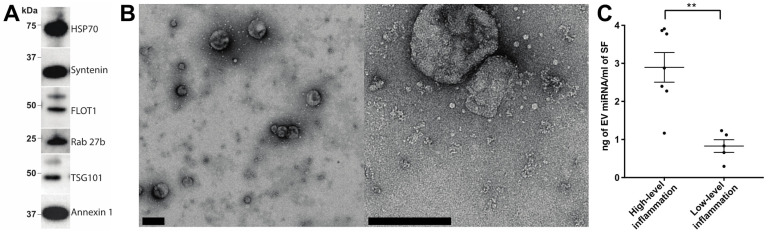
Assessment of synovial fluid EV enrichments prepared using size-exclusion chromatography. (**A**) Synovial fluid EV enrichments prepared by SEC were assessed for the presence of the canonical EV markers (HSP70, syntenin, FLOT1, Rab 27b, TSG101 and annexin 1) via Western blotting. (**B**) The quality of EV enrichments was further assessed by transmission electron microscopy (scale bars = 200 nm). (**C**) miRNA concentration per mL of rheumatoid arthritis synovial fluid in joints with high- or low-grade joint inflammation. Group means are indicated. Data analysed with Student’s *i*-test. Error bars = SEM. ** denotes *p*-value < 0.01.

**Figure 2 ijms-22-04910-f002:**
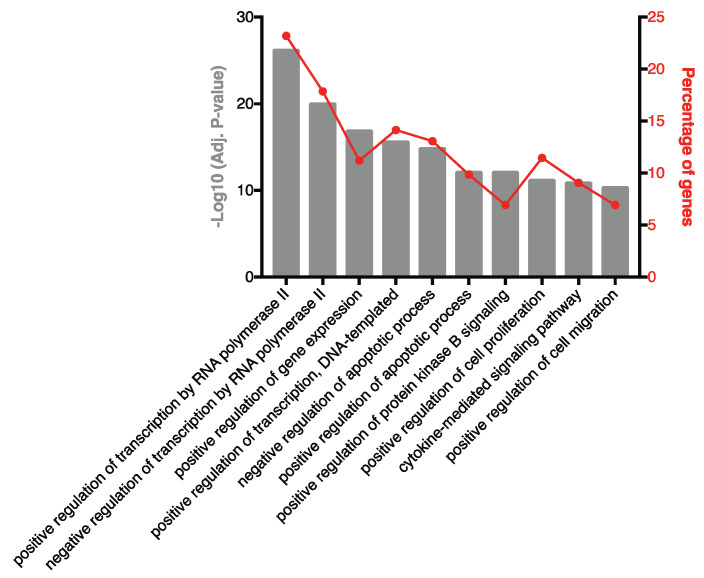
Biological processes associated with experimentally validated targets of highly ranked SF EV miRNA. Pathway analysis was performed on experimentally validated gene targets of the 10 highest-ranked miRNAs. The percentage of target genes associated with each biological process are indicated.

**Figure 3 ijms-22-04910-f003:**
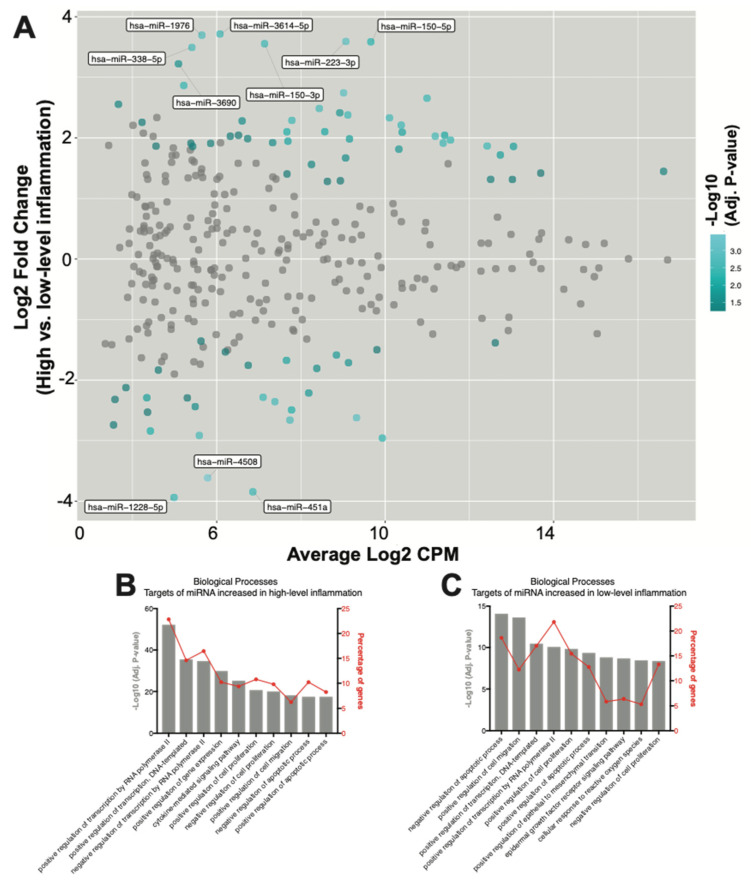
SF EV miRNAs have diverse immunoregulatory capacities. (**A**) MA plot of miRNA counts per million reads (CPM) vs. fold change. Differentially expressed miRNAs with an adjusted *p*-value < 0.05 are highlighted in blue. miRNAs with an adjusted *p*-value < 0.05 and a log2 fold change >3 are labelled. (**B**,**C**) Biological processes associated with target genes of the miRNAs significantly increased in joints with (**B**) high-, and (**C**) low-grade inflammation. The percentage of target genes associated with each biological process is indicated.

**Table 1 ijms-22-04910-t001:** Summary of patient details and clinical parameters.

	RA (High-Grade Inflammation)	RA (Low-Grade Inflammation)	*p*-Value
*n*	7	5	-
Age—mean (s.d.)	64.4 (11.8)	67.6 (10.0)	0.64
Sex—number of females/males	3/4	2/3	>0.99
White cell count—mean (s.d.) cells µL^−1^	8940 *	171.8	<0.001
Anti-citrullinated protein antibody (% positive)	71%	20%	0.24
Rheumatoid factor (% positive)	86%	60%	0.52
Disease Activity Score 28—median (range)	4.7 (3.31–5.41)	3.5 (2.74–5.0)	0.15
C-reactive protein—median (range) mg L^−1^	20 (6–164)	2 (1.4–2)	0.16

* *n* = 6. Sex, anti-citrullinated protein antibody and rheumatoid factor positivity analysed with Fisher’s exact test. All other parameters analysed with Student’s *t*-test.

**Table 2 ijms-22-04910-t002:** Highest-ranked SF EV miRNAs across the entire patient cohort.

Rank	miRNA	Average Expression (CPM)
1	hsa-miR-100-5p	132,441
2	hsa-miR-21-5p	179,929
3	hsa-miR-148a-3p	86,059
4	hsa-let-7a-5p	43,627
5	hsa-miR-92a-3p	45,881
6	hsa-let-7b-5p	31,280
7	hsa-miR-10b-5p	35,499
8	hsa-miR-99b-5p	31,569
9	hsa-miR-26a-5p	34,975
10	hsa-miR-99a-5p	45,879

**Table 3 ijms-22-04910-t003:** Common targets of the 10 highest-ranked EV miRNAs.

Target Gene ID	Target Gene Name	miRNA Regulators
IGF1R	Insulin like growth factor 1 receptor	hsa-let-7b-5p hsa-miR-100-5p hsa-miR-21-5p hsa-miR-99a-5p hsa-miR-99b-5p
CCND2	Cyclin D2	hsa-let-7a-3p hsa-let-7a-5p hsa-let-7b-5p hsa-miR-26a-5p
E2F2	E2F transcription factor 2	hsa-let-7a-3p hsa-let-7a-5p hsa-let-7b-5p hsa-miR-26a-5p
PTEN	Phosphatase and tensin homolog	hsa-miR-10b-5p hsa-miR-21-5p hsa-miR-26a-5p hsa-miR-92a-3p
STAT3	Signal transducer and activator of transcription 3	hsa-let-7a-5p hsa-miR-148a-3p hsa-miR-21-5p hsa-miR-92a-3p

**Table 4 ijms-22-04910-t004:** Differentially expressed miRNAs.

miRNA	Average Read Count (Log2 CPM)	Log2 Fold Change(High- vs. Low−Grade Inflammation)	Adjusted *p*-Value
hsa-miR-4508	5.78	−3.61	3.94 × 10^−4^
hsa-miR-223-3p	9.07	3.59	6.42 × 10^−4^
hsa-miR-3529-3p	9.01	2.74	6.42 × 10^−4^
hsa-miR-615-3p	9.32	−2.62	6.42 × 10^−4^
hsa-miR-1976	5.65	3.70	1.15 × 10^−3^
hsa-miR-543	7.74	−2.66	1.15 × 10^−3^
hsa-miR-338-5p	5.41	3.49	1.36 × 10^−3^
hsa-miR-146b-3p	8.43	2.49	1.36 × 10^−3^
hsa-miR-433-3p	7.38	−2.36	1.36 × 10^−3^
hsa-miR-485-3p	5.59	−2.91	1.36 × 10^−3^
hsa-miR-101-3p	10.10	2.33	1.52 × 10^−3^
hsa-miR-27a-5p	11.19	2.03	1.53 × 10^−3^
hsa-miR-361-3p	11.38	1.91	1.76 × 10^−3^
hsa-miR-3614-5p	6.08	3.72	1.86 × 10^−3^
hsa-miR-150-3p	7.14	3.55	1.89 × 10^−3^
hsa-miR-223-5p	9.11	2.38	1.89 × 10^−3^
hsa-miR-142-5p	7.78	2.29	1.89 × 10^−3^
hsa-miR-106b-3p	10.38	2.21	1.89 × 10^−3^
hsa-miR-28-3p	12.42	1.87	1.89 × 10^−3^
hsa-miR-455-5p	7.10	−2.28	1.89 × 10^−3^
hsa-miR-451a	6.86	−3.84	1.89 × 10^−3^
hsa-miR-143-3p	10.99	2.66	2.04 × 10^−3^
hsa-miR-1228-5p	4.99	−3.94	2.11 × 10^−3^
hsa-miR-30e-3p	11.55	1.96	2.25 × 10^−3^
hsa-miR-486-5p	9.93	−2.96	2.30 × 10^−3^
hsa-miR-1273h-3p	5.22	2.87	2.56 × 10^−3^
hsa-miR-150-5p	9.66	3.59	3.01 × 10^−3^
hsa-miR-378c	7.67	2.10	3.61 × 10^−3^
hsa-miR-92b-5p	7.78	−2.49	3.61 × 10^−3^
hsa-miR-4448	4.42	−2.84	3.61 × 10^−3^
hsa-miR-103b	8.57	2.10	4.76 × 10^−3^
hsa-miR-941	11.41	2.04	4.76 × 10^−3^
hsa-miR-103a-3p	7.69	1.95	5.24 × 10^−3^
hsa-miR-1246	10.40	2.09	5.64 × 10^−3^
hsa-miR-125b-1-3p	7.66	−1.67	5.64 × 10^−3^
hsa-miR-769-5p	9.15	1.98	6.68 × 10^−3^
hsa-miR-378a-3p	13.05	1.86	6.68 × 10^−3^
hsa-miR-140-3p	12.73	1.72	6.68 × 10^−3^
hsa-miR-214-5p	4.34	−2.29	6.68 × 10^−3^
hsa-miR-574-3p	8.84	−1.59	7.83 × 10^−3^
hsa-miR-1180-3p	8.18	−2.21	8.45 × 10^−3^
hsa-miR-155-5p	10.33	1.81	8.69 × 10^−3^
hsa-miR-629-5p	8.92	2.41	9.41 × 10^−3^
hsa-miR-328-3p	9.13	−1.71	1.05 × 10^−2^
hsa-miR-3690	5.09	3.22	1.18 × 10^−2^
hsa-miR-7704	8.38	−1.81	1.48 × 10^−2^
hsa-miR-221-5p	9.06	1.67	1.51 × 10^−2^
hsa-miR-486-3p	5.49	−2.44	1.51 × 10^−2^
hsa-miR-589-5p	6.60	2.28	1.56 × 10^−2^
hsa-miR-142-3p	7.33	1.92	1.56 × 10^−2^
hsa-miR-192-5p	8.25	1.56	1.98 × 10^−2^
hsa-miR-618	3.67	2.55	2.18 × 10^−2^
hsa-miR-21-5p	16.61	1.45	2.22 × 10^−2^
hsa-miR-21-3p	6.73	1.99	2.57 × 10^−2^
hsa-miR-345-5p	5.86	1.91	2.59 × 10^−2^
hsa-miR-214-3p	6.75	−1.76	2.59 × 10^−2^
hsa-miR-532-5p	12.50	1.31	2.70 × 10^−2^
hsa-miR-378e	5.43	1.86	2.86 × 10^−2^
hsa-miR-185-3p	6.32	2.03	3.05 × 10^−2^
hsa-miR-6787-3p	3.55	−2.74	3.07 × 10^−2^
hsa-miR-503-5p	4.23	2.26	3.11 × 10^−2^
hsa-miR-25-3p	13.02	1.31	3.11 × 10^−2^
hsa-miR-3120-5p	4.61	−1.83	3.24 × 10^−2^
hsa-miR-203b-5p	3.84	−2.12	3.35 × 10^−2^
hsa-miR-500a-3p	7.67	1.40	3.42 × 10^−2^
hsa-miR-7-5p	4.56	1.86	3.56 × 10^−2^
hsa-miR-3622a-5p	4.36	−2.53	3.56 × 10^−2^
hsa-miR-365a-5p	5.30	−2.29	3.76 × 10^−2^
hsa-miR-23b-3p	9.80	−1.50	3.85 × 10^−2^
hsa-miR-424-3p	6.52	2.04	4.29 × 10^−2^
hsa-miR-501-3p	8.94	1.29	4.30 × 10^−2^
hsa-miR-92b-3p	12.61	−1.38	4.37 × 10^−2^
hsa-miR-146b-5p	13.69	1.42	4.42 × 10^−2^
hsa-miR-185-5p	8.63	1.28	4.48 × 10^−2^
hsa-miR-23b-5p	6.21	−1.53	4.58 × 10^−2^
hsa-miR-378f	5.39	1.91	4.73 × 10^−2^
hsa-miR-27b-5p	5.63	−1.36	4.76 × 10^−2^
hsa-miR-1306-5p	3.58	−2.32	4.98 × 10^−2^

**Table 5 ijms-22-04910-t005:** Common targets of SF EV miRNAs found to be significantly enriched in the joints of RA patients with high-grade inflammation.

Target Gene ID	Target Gene Name	miRNA Regulators
IGF1R	Insulin like growth factor 1 receptor	hsa-miR-143-3p hsa-miR-150-3p hsa-miR-185-5p hsa-miR-21-5p hsa-miR-223-3p hsa-miR-223-5p hsa-miR-378a-3p hsa-miR-503-5p hsa-miR-7-5p
PTEN	Phosphatase and tensin homolog	hsa-miR-103a-3p hsa-miR-106b-3p hsa-miR-142-5p hsa-miR-155-5p hsa-miR-21-3p hsa-miR-21-5p hsa-miR-25-3p
VEGFA	Vascular endothelial growth factor A	hsa-miR-101-3p hsa-miR-150-5p hsa-miR-185-5p hsa-miR-21-5p hsa-miR-378a-3p hsa-miR-503-5p
BCL2	BCL2, apoptosis regulator	hsa-miR-143-3p hsa-miR-192-5p hsa-miR-21-5p hsa-miR-503-5p hsa-miR-7-5p
FBXW7	F-box and WD repeat domain containing 7	hsa-miR-155-5p hsa-miR-223-3p hsa-miR-223-5p hsa-miR-25-3p hsa-miR-503-5p
MYB	MYB proto-oncogene, transcription factor	hsa-miR-103a-3p hsa-miR-150-3p hsa-miR-150-5p hsa-miR-155-5p hsa-miR-503-5p
EGFR	Epidermal growth factor receptor	hsa-miR-146b-5p hsa-miR-21-5p hsa-miR-27a-5p hsa-miR-7-5p
RAC1	Rac family small GTPase 1	hsa-miR-101-3p hsa-miR-142-3p hsa-miR-142-5p hsa-miR-155-5p
TP53	Tumor protein p53	hsa-miR-150-3p hsa-miR-150-5p hsa-miR-25-3p hsa-miR-28-3p
ZEB1	Zinc finger E-box binding homeobox 1	hsa-miR-101-3p hsa-miR-142-5p hsa-miR-150-5p hsa-miR-223-3p

**Table 6 ijms-22-04910-t006:** Common targets of SF EV miRNAs found to be significantly enriched in the joints of RA patients with low-grade inflammation.

Target Gene ID	Target Gene Name	miRNA Regulators
PTEN	Phosphatase and tensin homolog	hsa-miR-214-3p hsa-miR-23b-3p hsa-miR-486-5p hsa-miR-92b-3p
TP53	Tumor protein p53	hsa-miR-125b-1-3p hsa-miR-214-3p hsa-miR-214-5p

## Data Availability

Raw fastq files have been uploaded to the Sequence Read Archive (PRJNA687167).

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
