# Peer review of "Extracellular Vesicles in Synovial Fluid from Rheumatoid Arthritis Patients Contain miRNAs with Capacity to Modulate Inflammation"

_ijms, 2021, doi:10.3390/ijms22094910_

Round 1

Reviewer 1 Report

In the present study, the authors profiled the Extracellular Vesicles's (EV) miRNAs within SF from a cohort of Rheumatoid Arthritis (RA) patients with either high or low-grade inflammation and also explored, bioinformatically, the mechanisms by which EV miRNAs might contribute to or protect against joint destruction in RA.

Their work has merit for publication after some minor revisions.

How did the authors come up with the common genes? did they use a Venn diagrammatic approach? please specify.

Were there common miRNAs between low- and high-grade RA? Add a table of common miRNAs expressed in both cases based on the top-ranked miRNAs. In addition, I would suggest to add a brief bioinformatics report on those miRNAs since it would be very interesting to see common mechanistic role of miRNAs in RA.

Some of the DE miRNAs reported with a p-value close to 0.05. Did the authors used some sort of FDR estimation? if no they should refer to it and if yes they should re-evaluate their results based on the calculated FDR and the p-values.

Finally, the authors should highlight their findings and mention possible clinical interventions derived from the detected miRNAs as well as how they can be used for better prognosis, diagnosis or therapy.

Author Response

In the present study, the authors profiled the Extracellular Vesicles's (EV) miRNAs within SF from a cohort of Rheumatoid Arthritis (RA) patients with either high or low-grade inflammation and also explored, bioinformatically, the mechanisms by which EV miRNAs might contribute to or protect against joint destruction in RA.

Their work has merit for publication after some minor revisions.

Thank you for the positive appraisal of our work.

How did the authors come up with the common genes? did they use a Venn diagrammatic approach? please specify.

We identified gene targets of miRNAs using miRTarBase, stated in the Materials and Methods; lines 353-4:

Gene targets of miRNA that have been experimentally observed by either western blot or reported assay were identified using miRTarBase 7.0 [18].

The gene targets are reported in Supplementary tables 5, 7 and 8 for the ten most abundant miRNAs, miRNAs enriched in high-grade inflammation and miRNAs enriched in low-grade inflammation, respectively.  From these tables, genes targeted by multiple miRNAs were identified.

Were there common miRNAs between low- and high-grade RA? Add a table of common miRNAs expressed in both cases based on the top-ranked miRNAs. In addition, I would suggest to add a brief bioinformatics report on those miRNAs since it would be very interesting to see common mechanistic role of miRNAs in RA.

We investigate common miRNAs likely involved in disease by ranking miRNA across all patients, as stated in Materials and Methods section: Ranking of prevalent miRNAs.  Targets and mechanistic roles of these miRNAs are considered and discussed throughout the manuscript.

Some of the DE miRNAs reported with a p-value close to 0.05. Did the authors used some sort of FDR estimation? if no they should refer to it and if yes they should re-evaluate their results based on the calculated FDR and the p-values.

As stated in the Materials & Methods on lines 339-40, P-values were adjusted using the Benjamini–Hochberg procedure.  DE data is presented in Figure 3A, Table 4 and Supplementary table 6 which all clearly specify an adjusted P value.

We agree that stating the P values when discussing miRNAs would be helpful and now include adjusted P values alongside fold changes.

Finally, the authors should highlight their findings and mention possible clinical interventions derived from the detected miRNAs as well as how they can be used for better prognosis, diagnosis or therapy.

Throughout the discussion we provide examples of how detected miRNAs might function in RA.  We also briefly consider therapeutic implications on lines 261-3:

Looking forward, the delivery of anti-inflammatory miRNAs via EVs and/or EV-mimic liposomes might present new therapeutic options for the treatment of RA.

Given this is a small descriptive study and considering current challenges with miRNA therapeutics, we feel that anything further would be overreach.

Reviewer 2 Report

this is a very novel and interesting study exploring the miRNA content of synovial exosomes in RA patients with either low or high grade inflammation.

Overall, the data a convincing, but several minors points need to be addressed:

1) in the patients caracteristics, I suggest adding th treatment, as this may (or may not) affect the interpretation of the data. 

2) could Ingenuity Pathy analysis be used, in order to avoid drawing conclusions from one algorithm of prediction only. Similarly, other target predictions programs need to be used, and only those common to 2 or 3 programs could be selected.

3) in table I , please explain why CRP level is not statistiacally different between high and low grade inflammation RA patients (the p=0.16 value is very surprising).

4) EV preparation. some more details are necessary regarding quality control of EVs. The refraction index? The size distribution ?

Author Response

this is a very novel and interesting study exploring the miRNA content of synovial exosomes in RA patients with either low or high grade inflammation.
Thank you.
Overall, the data a convincing, but several minors points need to be addressed:
1) in the patients caracteristics, I suggest adding th treatment, as this may (or may not) affect the interpretation of the data.
Treatment data is already included in Supplementary table 1 under the column, Medications.
2) could Ingenuity Pathy analysis be used, in order to avoid drawing conclusions from one algorithm of prediction only. Similarly, other target predictions programs need to be used, and only those common to 2 or 3 programs could be selected.
Respectfully, FunRich is a well-regarded pathway analysis tool for analysis of extracellular vesicles datasets. Our approach is line with methodological guidelines and standard pipelines (https://doi.org/10.1161/CIRCRESAHA.117.309417).
miRNA gene targets were identified based on strong experimental evidence as collated by miRTarBase. References for each miRNA:gene-target are provided in Supplementary tables 5, 7 and 8.
3) in table I , please explain why CRP level is not statistiacally different between high and low grade inflammation RA patients (the p=0.16 value is very surprising).
Individual CRP values are specified in Supplementary table 1. The small cohort and high inter-patient variability likely precluded statistical significance at an alpha of 0.05.
4) EV preparation. some more details are necessary regarding quality control of EVs. The refraction index? The size distribution ?
Thank you for this comment. We did not perform Nanoparticle Tracking Analysis on the patient cohort used in this study. However, we have previously published EV size distribution using a similar patient cohort (https://doi.org/10.1002/cti2.1185) and have comprehensively characterised size exclusion chromatography as a method for high quality EV enrichments (https://doi.org/10.1080/20013078.2018.1490145). We refer to these studies in the text.